# A Magnetic-Dependent Vibration Energy Harvester Based on the Tunable Point Defect in 2D Magneto-Elastic Phononic Crystals

**Tian Deng [1,2], Shunzu Zhang [1,2] and Yuanwen Gao [1,2,*]**

[1] Key Laboratory of Mechanics on Disaster and Environment in Western China attached to the Ministry of Education of China, Lanzhou University, Lanzhou 730000, China; dengt17@lzu.edu.cn (T.D.); zhangshz16@lzu.edu.cn (S.Z.)

[2] Department of Mechanics and Engineering Sciences, College of Civil Engineering and Mechanics, Lanzhou University, Lanzhou 730000, China

* Correspondence: ywgao@lzu.edu.cn; Tel.: +86-931-8914560; Fax: +86-931-8914560

**Abstract:** In this work, an innovative vibration energy harvester is designed by using the point defect effect of two-dimensional (2D) magneto-elastic phononic crystals (PCs) and the piezoelectric effect of piezoelectric material. A point defect is formed by removing the central Tenfenol-D rod to confine and enhance vibration energy into a spot, after which the vibration energy is electromechanically converted into electrical energy by attaching a piezoelectric patch into the area of the point defect. Numerical analysis of the point defect can be carried out by the finite element method in combination with the supercell technique. A 3D Zheng-Liu (Z-L) model which accurately describes the magneto-mechanical coupling constitutive behavior of magnetostrictive material is adopted to obtain variable band structures by applied magnetic field and pre-stress along the z direction. The piezoelectric material is utilized to predict the output voltage and power based on the capacity to convert vibration energy into electrical energy. For the proposed tunable vibration energy harvesting system, numerical results illuminate that band gaps (BGs) and defect bands of the in-plane mixed wave modes (*XY* modes) can be adjusted to a great extent by applied magnetic field and pre-stress, and thus a much larger range of vibration frequency and more broad-distributed energy can be obtained. The defect bands in the anti-plane wave mode (*Z* mode), however, have a slight change with applied magnetic field, which leads to a certain frequency range of energy harvesting. These results can provide guidance for the intelligent control of vibration insulation and the active design of continuous power supply for low power devices in engineering.

**Keywords:** vibration energy harvester; phononic crystal; defect bands; piezoelectric material; magnetostrictive material; output voltage and power

## 1. Introduction

With the ever-increasing development of self-powered wireless transmitters and embedded systems, the demands of independent power supply and extended lifespans become more and more intense [1,2]. In fact, energy harvesters of renewable and clear resources have attracted increasing attention of worldwide research communities with the increase of environmental issues caused by traditional resources. The various forms of those resources include sunlight, waste heat, flowing water, wind, and mechanical vibration, etc. [3–5]. Vibration energy especially, as a broad-distributed source, is the most prevalent within energy harvesting research [4], with numerous vibration energy generators of piezoelectric [6], electromagnetic [7], and electrostatic [8] conversion having been investigated. The piezoelectric generator, as one of the most effective collection devices, can harvest higher output

power owing to a better capability of electrical-mechanical coupling and higher strain for a given size. Considering the intrinsic advantage of high energy density in point defected phononic crystals (PCs), vibration energy can be accurately localized and enhanced at the point defect area, which can provide an excellent capability to achieve energy conversion through the direct piezoelectric effect.

As a typical composite material, PCs have different periodic structures, which can generate elastic/acoustic wave band gaps (BGs) where the wave propagation is forbidden in some ranges of wave frequency. With the introduction of some defects into perfect PCs, much attention has been concentrated on defect modes of PCs. Based on the characteristics of defect bands trapped in the BGs, where the elastic/acoustic wave can be localized and enhanced in the defect or propagate along the defected direction, defected PCs have extensive applications in engineering, such as being vibration isolators and noise suppressors [9], acoustic filters [10], and waveguides [11]. In recent years, a great deal of effort has been paid to obtain the formation mechanisms and influence factors of the point defected PCs. Khelif et al. [12,13] have theoretically and experimentally studied the characteristics of band structures and the localization effect of a 2D point-defected elastic PC. An accurate interferometric setup has been used by Romero-García et al. [14] for observing the symmetric and antisymmetric vibrational patterns in sonic crystals with double-point defects. Additionally, the bending wave propagation characteristics of a 2D point-defected PC thin plate have been discussed by Yao et al. [15]. Wu et al. [16] have reported the effects of superlattice configuration and defect shapes (square, circular, and rectangular) on defect bands of PCs. In fact, the control of defect bands for PCs with elastic materials has a significant dependence on not only the shapes of inclusions, lattice arrangement, and filling fraction, but also the different physical properties of components.

With the swift development of technology, considering the rich physics characteristics and extraordinary capabilities of smart materials and that their geometric and physical parameters can be changed greatly by external stimuli (magnetic field, pre-stress, temperature, and electric field, etc.), it is desirable and inevitable to construct dynamic tunable PCs with smart materials (magnetostrictive materials and piezoelectric materials, etc.) [17–21]. Regarding PCs with magnetostrictive material, researchers have experienced compelling interest in the modulation of elastic wave propagation and band structures in perfect magneto-elastic PCs by extrinsic motivation [22–25]. For example, Bou Matar et al. [22] have studied the wave propagation characteristics of 2D Terfenol-D/epoxy PCs, in which the BGs are tuned by adjusting the orientation and magnitude of the magnetic field. Considering the mechanical-magnetic coupling effect, Ding et al. [23] have presented longitudinal wave propagation characteristics in 1D Ni6/Terfenol-D PC rods. Zhang et al. [24] have found that the maximum width of BGs can reach an optimal orientation angle of 45° in the magnetic field for a magneto-elastic PC thin plate. However, for magneto-elastic PCs with defect modes, only a few studies have been reported in the existing literature. Gu and Jin [25] have explored the band structure of a 2D point-defected PC composed of Terfenol-D rods embedded in a polymethyl methacrylate (PMMA) matrix tuned by an applied magnetic field and pre-stress along the z-axis. The numerical results indicated that an applied suitable magnetic field can enlarge the first band gap (FBG) and capture a new band gap (NBG) in the in-plane modes (*XY* modes).

Owing to the wave localization and enhancement phenomena on account of point defect modes inside the BGs, vibration energy can be easily converted into electric energy by placing a ceramic piezoelectric patch within the defect state. There is considerable interest in the research of vibration energy harvesting by point-defected PCs [26–31]. Wu et al. [26] primarily used a polyvinylidene fluoride (PVDF) beam to harvest acoustic energy through a point-defected PC consisting of PMMA cylinders within the air background, and found that the maximum energy is collected at the resonance frequency of the point-defected PC and piezoelectric beam. In addition, when considering a 2D point-defected PC with a solid-solid system, Lv et al. [27] demonstrated experimentally that the output voltage and power harvesting efficiency are 421 and 177241 times larger than that in a rubber block, respectively. Based on the characteristics of wave focusing and energy localization, Carrara et al. [28] have demonstrated that acoustic wave energy harvesting can be produced by stub-plate acoustic

metamaterials. Yang et al. [29] studied a coupled resonance structure between two PC resonators in order to expand the acoustic wave localization and enhancement effects, with the experimental results revealing that the maximum pressure magnification of a coupled structure is three times larger than that of an individual PC. For the vibration energy harvester with point-defected PCs consisting of with elastic materials, the frequency of the defect band is a certain value. In fact, it is necessary to explore ways to enlarge the BGs and adjust the position of the defect bands, allowing the broad-distributed vibration energy to be converted into electrical energy. Hence, considering the magneto-electro-elastic coupling interaction and introducing smart materials (i.e., magnetostrictive material and piezoelectric material) into point-defected PCs is a good way to realize broad-frequency energy harvesting.

The aim of this paper is to present a way to modulate the range of vibration energy harvesting using magneto-elastic point-defected PCs with piezoelectric material, considering the tunability of the BGs and defect bands via an applied magnetic field and pre-stress. Two analytical approaches (band structure and vibration energy conversion) are proposed to quantitatively discuss magneto-electric conversion efficiency and further obtain the optimized output voltage and power of a vibration energy harvesting system. More specifically, a 3D nonlinear magneto-elastic coupling constitutive relationship in combination with the supercell technique is adopted to calculate band structure, and the direct piezoelectric effect is used to predict output voltage and power for energy harvesting by the finite element method (FEM) implemented by COMSOL Multiphysics 5.3a. [32]. These results provide a feasible way to broaden BGs and expand the frequency range of vibration energy harvesting simultaneously. This paper is organized as follows: In Section 2, the setup of the magneto-electro-elastic coupling theoretical model and calculation method are briefly presented. In Section 3, two schemes, i.e., the in-plane modes and the out-plane mode, are discussed. Finally, in Section 4 we give a conclusion.

## 2. Theoretical Model and Calculation Method

We consider a vibration energy harvester by inserting piezoelectric material into a 2D magneto-elastic PC with point defect from a $5 \times 5$ supercell. As shown in Figure 1a, the square lattice assumes that the lattice constant is $a$ and the radius of rod is $r$. The corresponding regions of rod, matrix and piezoelectric patch, indicated by $A$, $B$, and $C$, represent the magnetostrictive phase, elastic phase, and the piezoelectric phase. Figure 1b shows the first irreducible Brillouin zone of the supercell (the triangular area $\Gamma - X - M - \Gamma$) in the square lattice. A diagram of the magnetostrictive rod with applied pre-stress and magnetic field is shown in Figure 1c. The center of the rod is at the origin of the Cartesian coordinate system (o-xyz), with the x-y plane being located on the cross section of the rod and the z-axis being parallel to the length direction. Note that the pre-stress and magnetic field are applied along the z-axis of the Terfenol-D rod to tune the range of energy harvesting. The piezoelectric phase connecting to an electric circuit is shown in Figure 1d, where the piezoelectric patch is deformed by pressure difference $p$, so that the mechanical strain energy can be converted into electrical energy.

For the magnetostrictive phase, the Terfenol-D rod was chosen as the inclusion because it is widely used in high-precision actuators and smart devices due to their complex nonlinear magneto-mechanical coupling effect and sensitivity to external stimuli (magnetic field and pre-stress). The 3D Z-L nonlinear magneto-mechanical coupling constitutive equation is written as [33]:

$$\varepsilon_{ij} = \frac{1}{E}\Big[(1+v)\sigma_{ij} - v\sigma_{kk}\delta_{ij}\Big] + \frac{\lambda_s}{M_s^2}\Big[\frac{3}{2}M_iM_j - M_kM_k\Big(\frac{1}{2}\delta_{ij} + \widetilde{\sigma}_{ij}/\sigma_s\Big)\Big], \tag{1a}$$

$$H_k = \Big\{\frac{1}{kM}f^{-1}\Big(\frac{M}{M_s}\Big)\delta_{kl} - \frac{\lambda_s}{\mu_0 M_s^2}\Big[2\widetilde{\sigma}_{kl} - \Big(I_\sigma^2 - 3II_\sigma\Big)\delta_{kl}/\sigma_s\Big]\Big\}M_l. \tag{1b}$$

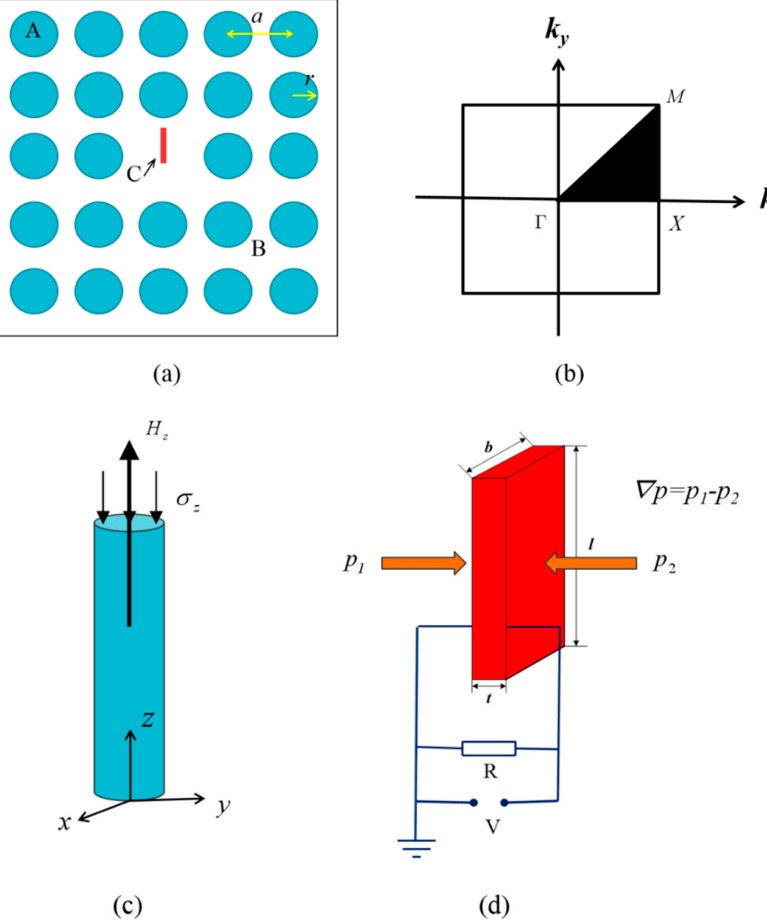

**Figure 1.** (**a**) Schematic of a 5 × 5 defected supercell of a 2D magneto-elastic PC with a piezoelectric patch. (**b**) The first irreducible Brillouin zone (triangular area $\Gamma - X - M - \Gamma$). (**c**) The Terfenol-D rod is affected by applied pre-stress and the magnetic field. (**d**) The structure and circuit connection of a piezoelectric patch.

In our study, by assuming that the Terfenol-D rod is infinite in magneto-elastic PCs, the demagnetization effect of the magnetostrictive material was able to be neglected [23]. Hence, the pre-stress and magnetic field were applied easily along the z-axis, that is, $\sigma_x = 0, \sigma_y = 0, H_x = 0, H_y = 0$. The corresponding constitutive relationship from Equation (1) is

$$
\begin{pmatrix}
\varepsilon_x \\
\varepsilon_y \\
\varepsilon_z \\
\gamma_{yz} \\
\gamma_{zx} \\
\gamma_{xy}
\end{pmatrix}
=
\begin{pmatrix}
1/E & -v/E & -v/E & 0 & 0 & 0 \\
-v/E & 1/E & -v/E & 0 & 0 & 0 \\
-v/E & -v/E & 1/E & 0 & 0 & 0 \\
0 & 0 & 0 & 1/G & 0 & 0 \\
0 & 0 & 0 & 0 & 1/G & 0 \\
0 & 0 & 0 & 0 & 0 & 1/G
\end{pmatrix}
\begin{pmatrix}
\sigma_x \\
\sigma_y \\
\sigma_z \\
\tau_{yz} \\
\tau_{zx} \\
\tau_{xy}
\end{pmatrix}
+
\frac{\lambda_s}{M_s^2}
\begin{pmatrix}
-1/2 - \widetilde{\sigma}_x/\sigma_s \\
-1/2 - \widetilde{\sigma}_y/\sigma_s \\
1 - \widetilde{\sigma}_z/\sigma_s \\
-2\widetilde{\tau}_{yz}/\sigma_s \\
-2\widetilde{\tau}_{zx}/\sigma_s \\
-2\widetilde{\tau}_{xy}/\sigma_s
\end{pmatrix}
M_z^2
\quad (2a)
$$

$$
H_z = \frac{1}{k} f^{-1}\left(\frac{M_z}{M_s}\right) - \frac{\lambda_s}{\mu_0 M_s^2}\left[2\widetilde{\sigma}_z - \left(I_\sigma^2 - 3II_\sigma\right)/\sigma_s\right] \cdot M_z
\quad (2b)
$$

where $\sigma_{ij}$ and $\varepsilon_{ij}$ represent the elastic stress and strain tensors, respectively, and $G = E/2(1+v)$ represents the shear modulus with $E$ and $v$ being Young's modulus and Poisson's ratio. $M = \sqrt{M_k M_k}$ is the magnetization intensity, and $\lambda_s$, $M_s$, and $\sigma_s$ represent, respectively, the saturation magnetostrictive coefficient, magnetization, and magnetostrictive stress. $H_z$ and $M_z$ represent the magnetic field and magnetization intensity along the z-axis, respectively. $k = 3\chi_m/M_s$ is the relaxation factor, where $\chi_m$ is the susceptibility in the initial linear region. For $I_\sigma^2 - 3II_\sigma = 2\widetilde{\sigma}_{ij}\widetilde{\sigma}_{ij}/3$, $\widetilde{\sigma}_{ij} = 3\sigma_{ij}/2 - \sigma_{kk}\delta_{ij}/2$ represents 3/2

times the deviatoric stress $\sigma_{ij}$, and $\delta_{ij}$ is the Kronecker delta. $\mu_0 = 4\pi \times 10^{-7}$ (H/m) represents the vacuum permeability. The nonlinear scalar function $f(x) = \coth(x) - 1/x$ represents the Langevin function, which is based on Boltzmann statistics and can give a better calculation for the magnetization curve [33].

It is difficult to directly incorporate the nonlinear constitutive equations into the mechanical equations because the equations contain inverse functions and implicit expressions. For simplicity, we can rewrite the above nonlinear constitutive equations of magnetostrictive material as the general form of the linear-like constitutive equations with variable equivalent coefficients, which are widely used in much magneto-elastic PCs research and reflect the nonlinear constitutive relations effectively in macroscopic scenarios [17–19].

$$\varepsilon_{kl}^m = s_{ijkl}^m(\boldsymbol{\sigma}, \boldsymbol{H})\sigma_{ij}^m - d_{mkl}^m(\boldsymbol{\sigma}, \boldsymbol{H})^T H_m^m, \tag{3a}$$

$$B_n^m = d_{nij}^m(\boldsymbol{\sigma}, \boldsymbol{H})\sigma_{ij}^m + \mu_{nm}^m(\boldsymbol{\sigma}, \boldsymbol{H})H_m^m, \tag{3b}$$

where the right superscript $m$ denotes the magnetostrictive phase. $B_n^m = \mu_0^m(H_m^m + H_n^m)$ represents the magnetic induction vector, and $s_{ijkl}^m(\boldsymbol{\sigma}, \boldsymbol{H})$, $d_{mkl}^m(\boldsymbol{\sigma}, \boldsymbol{H})$, and $\mu_{nm}^m(\sigma, H)$ represent, respectively, the flexibility matrix, the piezomagnetic coupling matrix and the magnetic permeability matrix, which are the functions of the magnetic field and pre-stress. $(.)^T$ denotes the transposition of the matrix. $\sigma$ and $H$ represent, respectively, the stress and magnetic field intensity vectors, which are the independent variables. The effective material coefficients and exact expressions can be found in Ref. [25].

For the piezoelectric phase, piezoelectric material as the vibration energy generator has received wide attention in research because of its high efficiency of electrical-mechanical conversion and high output voltage. The 31 piezoelectric mode is adopted for electricity generation of piezoelectric patches, and the pressure difference $p$ applied across the two sides of the patch acts as the external force to drive the vibration of the piezoelectric patch, which converts the vibration energy into electrical energy. Because the load resistance of the external loading circuit has a significant effect on the piezoelectric energy harvesting, the electrical impedance is in series at the two sides of the piezoelectric patch to find the optimal output voltage and power [31]. In the development of the generator structure for the piezoelectric patch, the established 3D linear piezoelectric constitutive equation in reduced-matrix form is [34]

$$\begin{bmatrix} D^p \\ \sigma^p \end{bmatrix} = \begin{bmatrix} \varepsilon^s & e^p \\ -(e^p)^T & c^p \end{bmatrix} \begin{bmatrix} E^p \\ \varepsilon^p \end{bmatrix} \tag{4}$$

where the right superscript $p$ denotes the piezoelectric phase. $\sigma^p$ and $D^p$ are the stress and electric displacement vectors, respectively. $\varepsilon^p$ and $E^p$ are the strain and electric field vectors, respectively. $c^p$, $e^p$, and $\varepsilon^s$ are the elastic coefficient at constant electric field, piezoelectric stress coefficient, and dielectric constant at constant stress field, respectively. $(.)^T$ denotes the transposition of the matrix.

According to the [31], both output voltage and power are functions of the external load resistance. It can be found that the maximum value of output power can be reached when the optimal loading resistance is yielded as

$$R_{opt} = \frac{1}{\omega C_p} \frac{2\zeta}{\sqrt{4\zeta^2 + k^4}} \tag{5}$$

where $\omega$, $\zeta$, $k$, and $C_p$ are the forcing frequency of the piezoelectric patch, damping radio, piezoelectric coupling coefficient, and capacitance of the piezoelectric patch, respectively.

In order to obtain the band structure of point defected PCs, a periodic boundary condition based on the Bloch-Floquet theorem was applied on the interfaces of both the x and y directions between the adjacent supercell systems.

$$u_i = (x + a_1, y + a_1) = u_i(x, y)e^{i(k_x a_1 + k_y a_1)}, (i = x, y, z) \tag{6}$$

where $u_i$ represents the elastic displacement vector, $a_1 = 5 \times a$ denotes the lattice constant of the supercell, and $k_x$ and $k_y$ are the Bloch vectors. The supercell can be meshed by the quadratic Lagrange

triangular element, and a group of eigenvalues and eigenmodes are obtained by scanning the Bloch wave vector $k$ from the first irreducible Brillouin zone along the path of $\Gamma - X - M - \Gamma$. Hence, the defect bands of magnetic-elastic PCs can be obtained by applying different magnetic fields and pre-stress.

In this study, the FEM is utilized to analyze and calculate the system of vibration energy harvesting based on magneto-elastic PCs with point defects. The structural mechanics and AC/DC modules are used to model the coupling problems of the mechanical, magnetic, and electrostatic fields and to estimate the output voltage and power from the piezoelectric generators. The relationship between the magnetic-electro-mechanical coupling is displayed in Figure 2. The structural mechanics module is used to calculate the band structure of the $5 \times 5$ supercell with a point defect under a different magnetic field and pre-stress. The Partial Differential Equation (PDE) module is adopted to solve the implicit function problem in the effective material parameters of magnetostrictive material. Using the AC/DC module, the piezoelectric patch connecting to an electric circuit can be utilized to convert the mechanical vibration into electrical energy, and then the series resistance is added in the circuit module to obtain the maximum output power. Thus, the output voltage and power can be harvested from point defected magneto-elastic PCs with piezoelectric material.

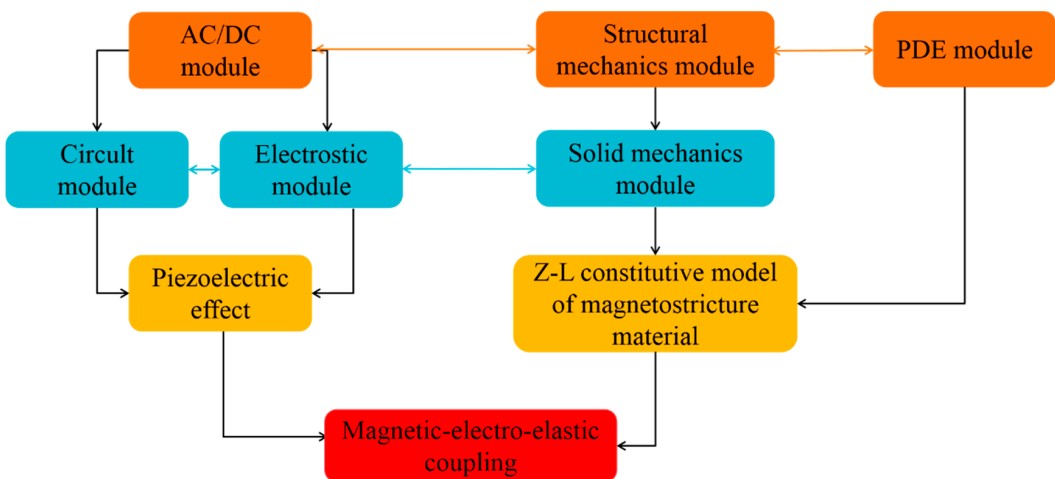

**Figure 2.** The relationships between the magnetic-electro-mechanical coupling of each module.

## 3. Numerical Results and Discussion

### 3.1. Verification of Point-Defect Bands

In this section, we adequately investigate the tunable vibration energy harvesting system of a 2D magnetic-elastic (M-E) PC including point defect modes (in-plane modes and out-plane modes) under different magnetic and stress fields. It is noteworthy that the magneto-elastic system in this paper can be degenerated directly to the study of a conventional point-defected PC with elastic material by excluding the magnetostrictive material. The band structure of the point-defected PC consisting of a steel rod and rubber matrix is calculated in Figure 3 and compared with the results of Ref. [35] in order to validate the accuracy of the proposed model. The results show the BG occurred in the frequency range of 564–1038 Hz and two defect bands appeared at 648.2 and 648.5 Hz in the frequency range of 0–1100 Hz. It can be seen that the results of the PC with elastic material (black lines) are consistent with the previous results (red dots) qualitatively and quantitatively, which verifies the accuracy of the current numerical model. As a specific case, the M-E PC material parameters of Terfenol-D [36] and PMMA [37] are listed in Table 1. The physical parameters of the piezoelectric patch (PZT-5A) [38] are shown in Table 2. The length ($l$), width ($b$), and thickness ($t$) of the piezoelectric patch are set as 20 mm, 7 mm, and 0.7 mm, respectively. The damping ratio $\xi$ is chosen to be 0.025. The lattice constant $a$ and the radius of scatter cylinder $r$ are 15.5 mm and 5.5 mm, meaning the corresponding filling fraction is $f = \pi r^2 / a^2 = 39.6\%$.

**Table 1.** Material parameters of Terfenol-D [36] and polymethyl methacrylate (PMMA) [37].

| Materials | $\rho\,(\mathrm{kg/m^3})$ | $E(\mathrm{GPa})$ | $\upsilon$ | $C_{11}\,(\mathrm{GPa})$ | $C_{44}\,(\mathrm{GPa})$ | $\lambda_s\,(\mathrm{ppm})$ | $\chi_m$ | $\sigma_s\,(\mathrm{GPa})$ | $\mu_0 M_s\,(\mathrm{T})$ |
|---|---|---|---|---|---|---|---|---|---|
| Terfenol-D | 9200 | 60 | 0.3 | — | — | 1950 | 20.4 | 200 | 0.96 |
| PMMA | 1200 | — | — | 7.11 | 2.03 | — | — | — | — |

**Table 2.** Physical parameters of the piezoelectric patch (PZT-5A) [38].

| Physical Parameters | Elastic Coefficient (GPa) | | | | | Piezoelectric Coefficient ($10^{-12}$ C/m$^2$) | | | Dielectric Constant ($10^{-9}$ F/m) | |
|---|---|---|---|---|---|---|---|---|---|---|
| | $c_{11}^p$ | $c_{12}^p$ | $c_{13}^p$ | $c_{33}^p$ | $c_{44}^p$ | $e_{31}^p$ | $e_{33}^p$ | $e_{15}^p$ | $\varepsilon_{11}^s$ | $\varepsilon_{33}^s$ |
| PZT-5A | 121 | 75.40 | 75.20 | 111 | 21.1 | -5.4 | 15.8 | 12.3 | 8.107 | 7.346 |

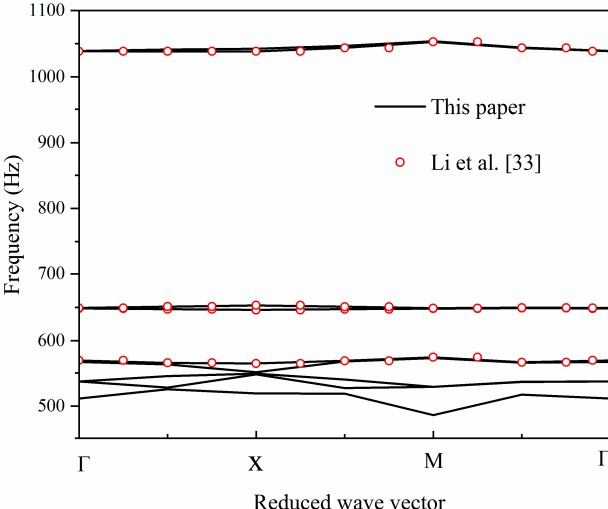

**Figure 3.** Comparisons of elastic PC of steel/rubber between the results of this paper (black lines) and those of Ref. [35] (red dots).

### 3.2. Output Voltage and Power for in-Plane Modes (XY Modes)

The effects of the magnetic field, pre-stress, and the piezoelectric patch on the band structure (*XY* modes) of the M-E PC with point defect are depicted in Figure 4. The transverse coordinate represents the reduced wave vector and the vertical coordinate represents the normalized frequency ($\omega a/2\pi c_t$). $\omega$ represents the frequency and $c_t$ = 1300 m/s represents the transverse wave velocity of PMMA. It can be observed in Figure 4a that one BG appears in the band structure and no defect band exists in the PC without the point defect mode being $\sigma_z$ = 0 MPa, $H_z$ = 0 kOe. It can be seen from Figure 4b that the range of the first band gap (FBG) is 0.545006–1.050319, where three defect bands (original defect bands) are observed when $\sigma_z$ = 0 MPa, $H_z$ = 0 kOe. Figure 4c shows the band structure of the M-E PC with a point defect computed by considering the structural effect of the piezoelectric patch. By comparing Figure 4b,c, we can confirm that the piezoelectric patch has no obvious effect on the BGs and defect bands. When the stress is set to 0 MPa, and the magnetic field changes from 0 to 1 kOe, as shown Figure 4d, the lower edge of the FBG is slightly changed, while the upper edge FBG increases to 1.177729, and three new defect bands appear above the original defect bands, meaning that the width of the FBG increases by 0.127410. Note that a new band gap (NBG) appears in range 1.337568–1.450522, in which the four defect bands are trapped. When $H_z$ = 1 kOe, the compressive pre-stress changes from 0 to 20 MPa, as shown Figure 4e, and the range of the FBG is 0.545954–1.061739. In comparison with Figure 4d, the upper edge of the FBG decreases, and new defect bands of FBG and NBG are closed. Some displacement fields of the eigenmodes (NBG1$^{\text{st}}$, NBG2$^{\text{nd}}$, NBG3$^{\text{rd}}$, and NBG4$^{\text{th}}$) display the vibration confinement and the double degenerate modes on the point defect states of the

M point, as shown in Figure 5, when the $\sigma_z$ = 0 MPa and $H_z$ = 1 kOe. Hence, it can be concluded that under the action of the magnetic field and pre-stress, the material parameters of Terfenol-D are changed, which leads to change in the BGs and the appearance of new defect bands; similar results can be found in Ref. [25]. In the following calculation, we will pay more attention to the investigation of the tunable vibration energy harvester used by the M-E PC with point defect.

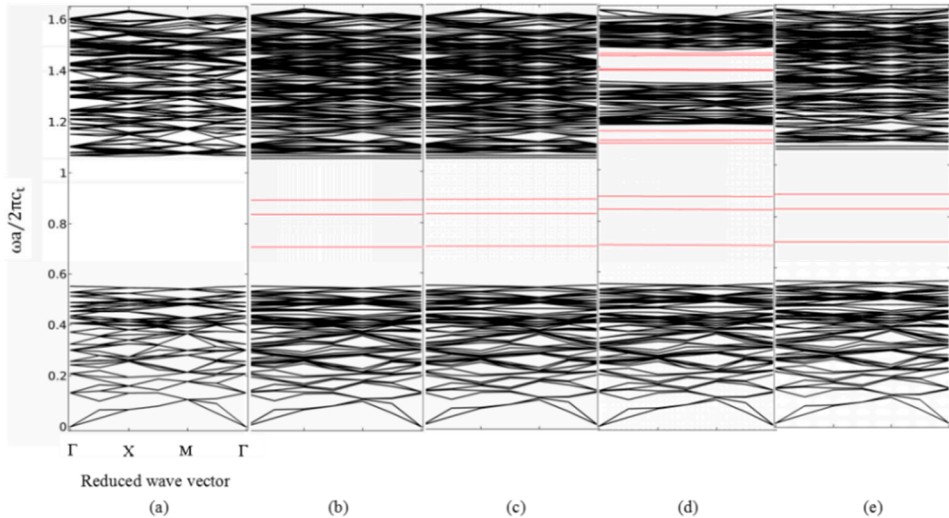

**Figure 4.** The band structure ($XY$ modes) of a magnetic-elastic (M-E) PC with point defect under different magnetic fields and pre-stress values. (**a**) perfected M-E PC at $\sigma_z$ = 0 MPa, $H_z$ = 0 kOe. (**b**) $\sigma_z$ = 0 MPa, $H_z$ = 0 kOe without PZT-5A. (**c**) $\sigma_z$ = 0 MPa, $H_z$ = 0 kOe with PZT-5A. (**d**) $\sigma_z$ = 0 MPa, $H_z$ = 1 kOe with PZT-5A. (**e**) $\sigma_z$ = −20 MPa, $H_z$ = 1 kOe with PZT-5A.

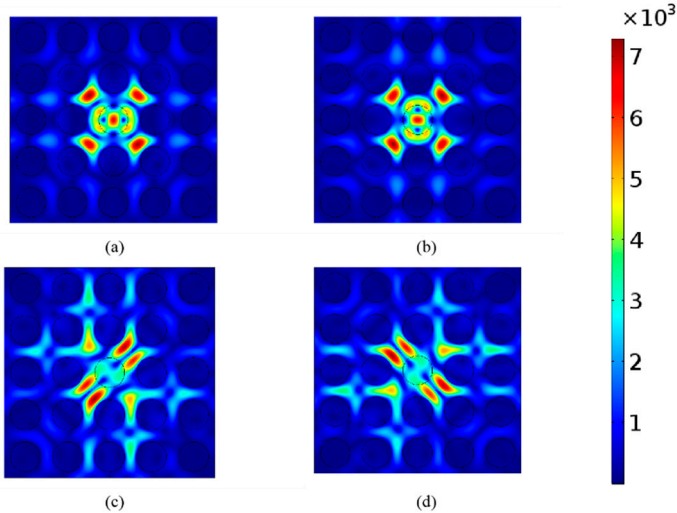

**Figure 5.** Displacement distributions of the point defect at the M point for $\sigma_z$ = 0 MPa, $H_z$ = 1 kOe. (**a**) NBG1$^{\text{st}}$ is at $\omega a/2\pi c_t$ = 1.385799, (**b**) NBG2$^{\text{nd}}$ is at $\omega a/2\pi c_t$ = 1.388351, (**c**) NBG3$^{\text{rd}}$ is at $\omega a/2\pi c_t$ = 1.439993, and (**d**) NBG4$^{\text{th}}$ is at $\omega a/2\pi c_t$ = 1.447252.

Figure 6 displays the variation of BGs and defect bands with the magnetic field in an M-E point-defected PC as $\sigma_z$ = 0 MPa and $\sigma_z$ = −20 MPa, respectively. The lower and upper edge of the FBG represent the start and cut off frequency, respectively, of the first band gap. The six defect bands trapped in the FBG are expressed as 1$^{\text{st}}$, 2$^{\text{nd}}$, 3$^{\text{rd}}$, 4$^{\text{th}}$, 5$^{\text{th}}$, and 6$^{\text{th}}$, respectively. The NBG and those corresponding to defect bands have the same representation. When the magnetic field increases from 0 to 4 kOe and $\sigma_z$ = 0 MPa, Figure 6a shows that the position of the lower edge of the FBG and the original defect bands of the FBG (FBG1$^{\text{st}}$, FBG2$^{\text{nd}}$, and FBG3$^{\text{rd}}$) stay nearly unchanged, and the upper

edge of the FBG increases and tends to a stable state as $H_z$ increases. It is noteworthy that the new defect bands of the FBG are opened at $H_z = 0.3$ kOe, and then gradually increase and tend to a fixed value with the increase of $H_z$. Moreover, the NBG and corresponding defect bands are opened up when $H_z = 0.5$ kOe, and the width and more defect bands of NBG increase with the rise of $H_z$. It is interesting that the defect bands in the frequency ranges 1.07–1.19 and 1.31–1.47 gradually separate and become flatter. When increasing the compressive pre-stress to 20 MPa, by comparing Figure 6a,b, it can be seen that the new defect bands of the FBG and NBG open in the higher magnetic field (1.1 kOe), and the edges and width of all these BGs and defect bands increase gradually and finally reach a certain constant at the saturated magnetic field. These results show that not only does the magnetic field have a great effect on the BGs and defect bands, but that pre-stress also has a significant effect on them. From the viewpoint of the magnetic domain, when $H_z$ is in the low and intermediate field, the physics mechanism for these changes is that the magnetic domain rotation of the Terfenol-D rod is along the direction of easy magnetization with the rise of the magnetic field, resulting in the expansion range of the FBG and the generation of the NBG. Then, the magnetization of the magnetostrictive material is up to saturation, resulting in the BGs and corresponding defect bands remaining constant when $H_z$ reaches the high field. In addition, when $H_z$ is in the low and intermediate field, the applied pre-stress can make the magnetic domain rotate toward the direction of hard magnetization. In order to achieve the same magnetization intensity under larger pre-stress, a larger magnetic field must be applied. It is difficult for pre-stress to affect the saturation magnetization when $H_z$ reaches the high field.

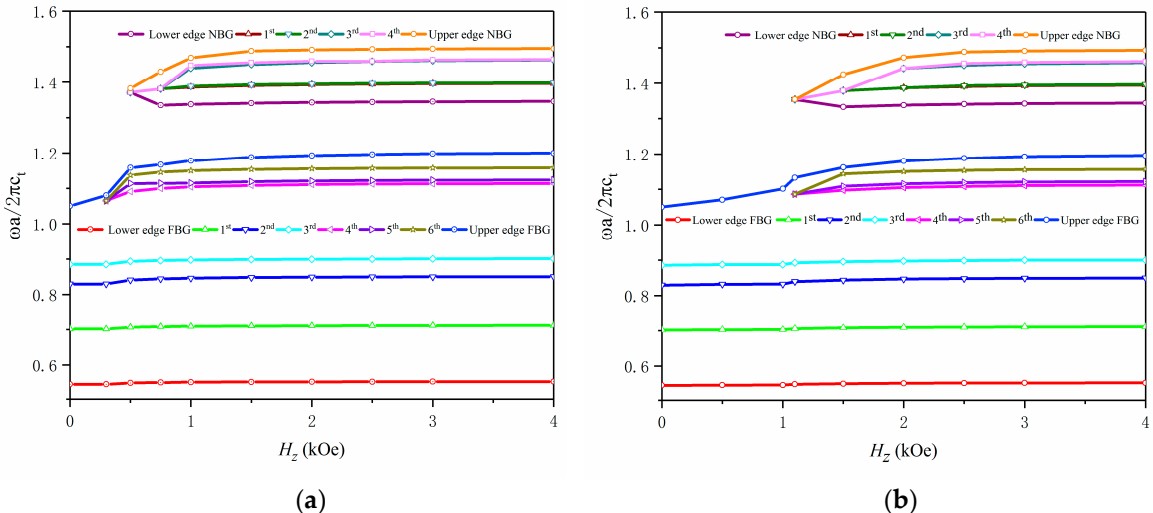

**Figure 6.** The variation in defect band frequency (*XY* modes) as a function of magnetic field. (**a**) $\sigma_z = 0$ MPa and (**b**) $\sigma_z = -20$ MPa. Legend: NBG, new band gap; FBG, first band gap.

Figure 7 presents the output voltage of defect bands in the FBG and NBG as a function of magnetic field in which $\sigma_z = 0$ MPa and R = 20 kΩ. Here, the output voltages at the defect bands of the FBG are expressed as FBG1[st], FBG2[nd], FBG3[rd], FBG4[th], FBG5[th], and FBG6[th], respectively. Output voltages at the defect bands of NBG are expressed as NBG1[st], NBG2[nd], NBG3[rd], and NBG4[th], respectively. The vibration energy can be converted into electrical energy by applied acceleration of 1 m/s² on the left-hand side of the supercell, incident from the x-axis. It can be seen from Figure 7a that the output voltage of the FBG1[st] is 11 mV and that the output voltages of the FBG4[th], FBG5[th], and FBG6[th] begin to appear simultaneously in $H_z = 0.3$ kOe. The output voltages of FBG2[nd] and FBG3[rd] become constant at 0.2 mV with the increases of the magnetic field. The output voltages of FBG4[th] are approximately in the range of 1.1–1.5 mV when $H_z$ is in the low and intermediate field and tend to 1.1 mV in the high field. The output voltages of the FBG5[th] and FBG6[th] tend to 0.2 mV with the rise of the magnetic field. It is shown in Figure 7b that the output voltage of the defect bands in the NBG are opened up with the magnetic field at 0.5 kOe, and the output voltages of NBG1[st], NBG2[nd], and NBG3[rd] rapidly decrease

with the magnetic field in the low and intermediate field. The output voltages of NBG1st and NBG2nd reach a steady state of about 5 mV, and those of NBG3rd reach a plateau of about 10 mV in the high field. The output voltage of NBG4th gradually increases to a maximum value of 52 mV as the magnetic field increases. Hence, we can conclude that the magnetic field changes the displacement field distributions of the defect bands, which leads to variation of the collected voltage in the magnetic field in the low and intermediate field. However, the displacement field of the defect bands is unchanged at the high field, which results in the stability of the output voltage. It can be concluded that the higher output voltage of 52 mV can be collected in the NBG compared to the FBG. In order to find the larger output voltage and power, in the following work we focus on the vibration energy harvesting range of the NBG.

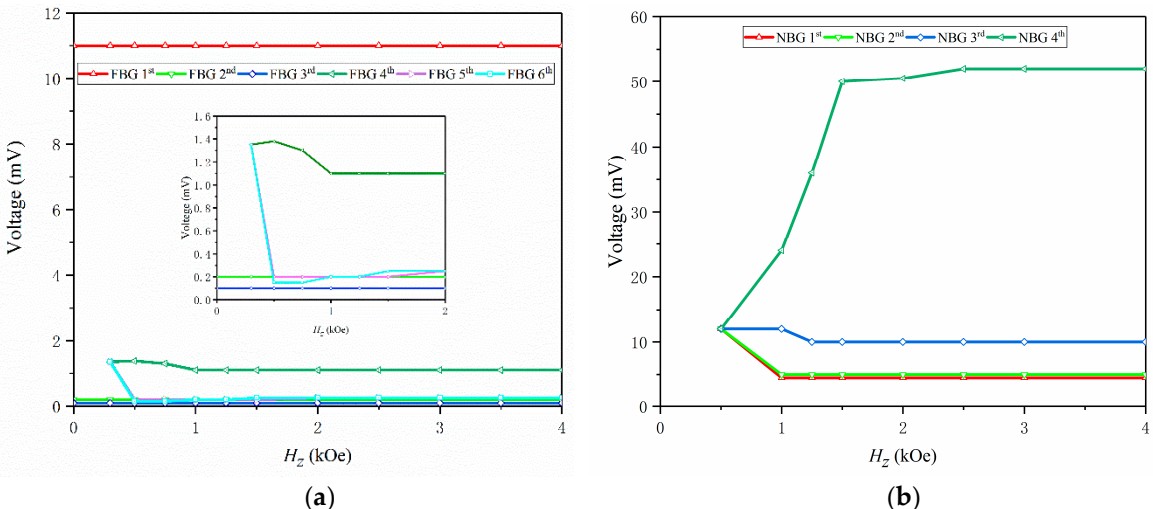

**Figure 7.** The output voltage of defect bands (*XY* modes) versus the magnetic field when $\sigma_z = 0$ MPa, R = 20 kΩ. (**a**) The output voltage of the FBG. (**b**) The output voltage of the NBG.

Figure 8 illustrates the dependence of the output voltage of the NBG on the frequency under different magnetic fields and pre-stress values. The effect of the magnetic field on the output voltage for the defect bands is illustrated in Figure 8a. The voltage can hardly be collected at $H_z = 0$ kOe. The output voltage has three peak values when $H_z = 1.1$ kOe, with the corresponding frequencies being 1.388752, 1.445999, and 1.456722, respectively. As a result of the frequencies of NBG1st and NBG2nd being very close, there is an output voltage peak in the voltage-frequency curve, and other frequencies corresponding to the peak voltage are consistent with the frequencies of the defect bands. The frequencies of the three peaks output voltage for $H_z = 2$ kOe can be found at 1.392721, 1.455566, and 1.462921, respectively, which also agree well with the defect band frequencies in the NBG. The output voltage versus the frequency of the defect bands in NBG at $\sigma_z = -20$ MPa is shown in Figure 8b. The voltage can hardly be collected at $H_z = 0$ kOe, the output voltage only has one peak with the frequency of 1.412125 at $H_z = 1.1$ kOe, and the output voltage has three peak values at $H_z = 2$ kOe, with the corresponding frequencies being 1.387582, 1.444213, and 1.448955, respectively, in agreement with the frequencies of the defect bands. In order to express more clearly and intuitively the effect of pre-stress on the output voltage, a curve of the output voltage with frequency under different pre-stress values at $H_z = 2.5$ kOe is given in Figure 9. The results show that the frequency corresponding to the peak voltage moves to the lower frequency region and that the highest peak voltage of the NBG gradually decreases with the incensement of compressive pre-stress. On the other hand, these phenomena show that the localized and enhanced characteristics of point-defected PC can be used to harvest the highest voltage of the piezoelectric patch.

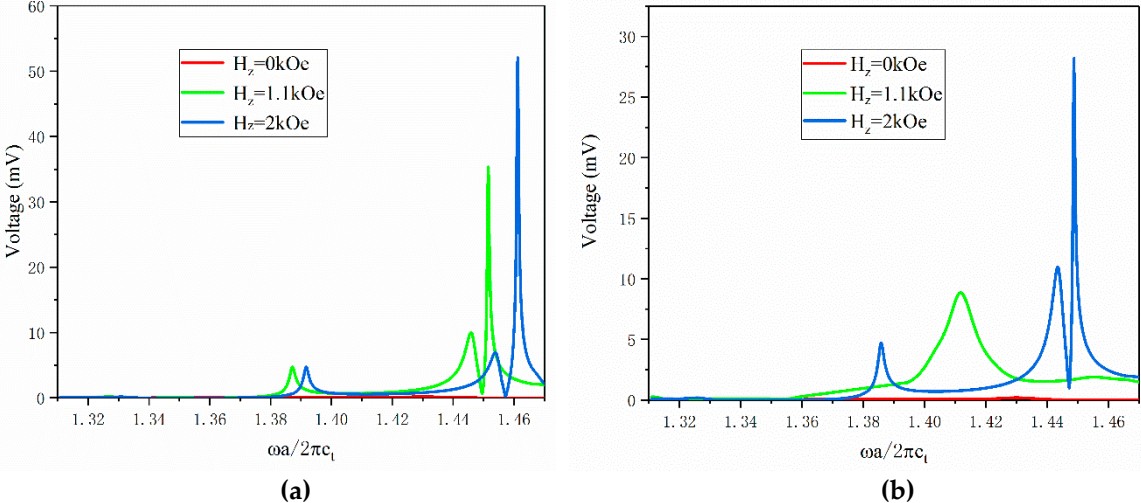

**Figure 8.** Output voltage versus the normalized frequency in NBG with R = 20 kΩ. (**a**) At $\sigma_z$ = 0 MPa. (**b**) At $\sigma_z$ = −20 MPa.

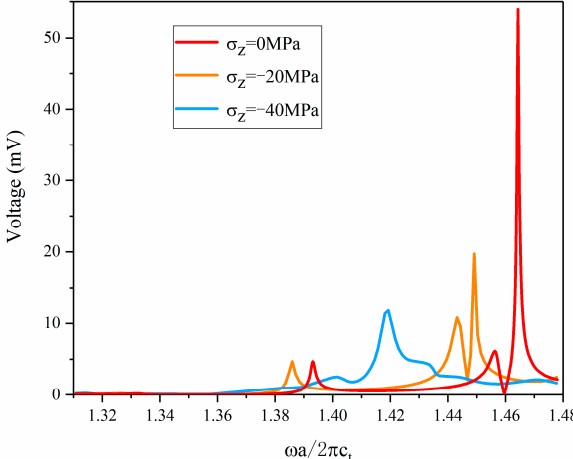

**Figure 9.** The influence of pre-stress on the output voltage in the NBG when the magnetic field is 2.5 kOe.

Figures 9 and 10 show the output voltage spectrum as a function of frequency under different magnetic fields and pre-stress values, respectively. Hence, quality factor can be introduced in this study, where the quality factor is defined by the ratio of the frequency in the defect band over the bandwidth between the half-voltage point ($\Delta\omega$) [39]. Note that the frequency of defect band corresponds to the highest peak voltage and the $\Delta\omega$ can be easily obtained from the output voltage spectrum to calculate the quality factor. Hence, the quality factor of the M-E PC with point defect under different magnetic fields and pre-stress values is presented in Table 3. We can see that when the pre-stresses are 0 and −20 MPa, the quality factor decreases gradually with a rise in magnetic field, which is opposite to the highest peak voltage. In particular, the quality factor reaches the lowest value because of the lower frequency corresponding to a higher bandwidth at $H_z$ = 1.1 kOe, $\sigma_z$ = −20 MPa. Moreover, under the high magnetic field, the quality factor increases gradually with the increases of pre-stress.

Figure 10 shows the output voltage and power as a function of load resistance at defect band frequencies of the NBG (NBG1[st], NBG2[nd], NBG3[rd], and NBG4[th]) when $\sigma_z$ = 0 MPa, $H_z$ = 1 kOe. It is shown in Figure 10a that the output voltage at defect band frequencies increases rapidly to reach a fixed value with an increasing of load resistance initially. It can be easily seen in Figure 10b that the output power at defect band frequencies gradually increases with a rise of load resistance until an extreme value, followed by a decrease and finally a stabilization. It is demonstrated that there is an

optimal load resistance of R = 11 kΩ corresponding to a maximum output power of *P* = 17 nW when the frequency is in NBG4$^{th}$. Note that the optimal load resistance agrees with the result of Equation (5).

**Table 3.** The achieved $\Delta\omega$ and quality factor results from different magnetic fields and pre-stress values.

| Magnetic Field $H_z$ (kOe) | Pre-Stress $\sigma_z$ (MPa) | Frequency of Defect Band (Hz) | Bandwidth ($\Delta\omega$) | Quality Factor |
|---|---|---|---|---|
| 1.1 | 0 | 121,410 | 809 | 150.074 |
| 2 | 0 | 122,384 | 960 | 127.483 |
| 2.5 | 0 | 122,579 | 1215 | 100.888 |
| 1.1 | −20 | 118,401 | 1838 | 64.418 |
| 2 | −20 | 120,948 | 739 | 163.664 |
| 2.5 | −20 | 122,078 | 920 | 132.693 |

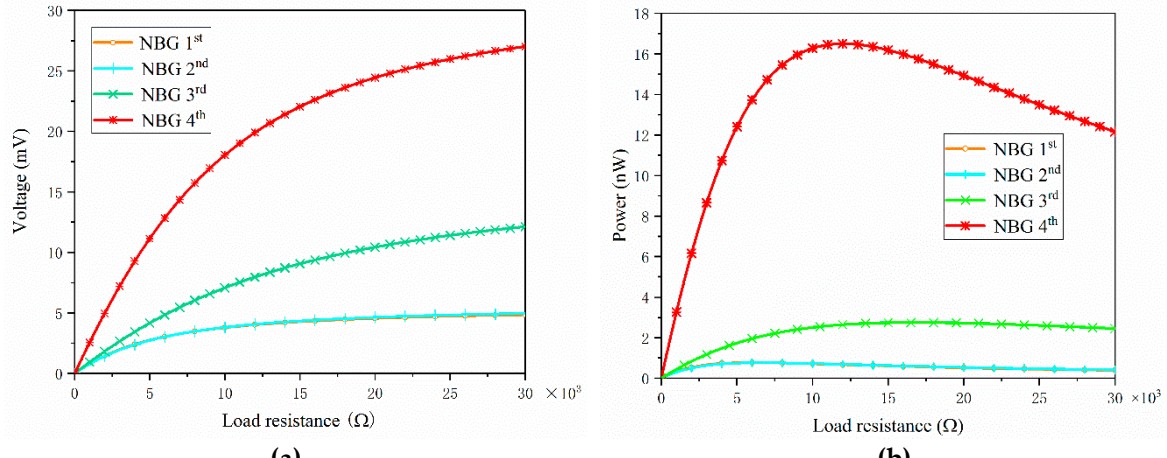

**Figure 10.** The output voltage (**a**) and power (**b**) versus the load resistance at the frequencies of the defect bands in the NBG at $\sigma_z$ = 0 MPa, $H_z$ = 1 kOe.

Figure 11 illustrates that the output voltage is a sinusoidal function of time at defect bands of the NBG when $\sigma_z$ = 0 MPa, $H_z$ = 1 kOe, R = 20 kΩ. The period (*T*) of the defect band frequency can be represented by $T = 1/\omega$. It can be verified that the values of *T* are 8.61 µs, 8.60 µs, 8.27 µs, and 8.24 µs, which correspond to the frequencies of the four defect bands in the NBG, the 1$^{st}$, 2$^{nd}$, 3$^{rd}$ and 4$^{th}$, respectively. It is shown that the corresponding amplitudes of the output voltage at the four defect band frequencies are 4.5 mV, 4.6 mV, 10.5 mV, and 22.5 mV, respectively, which is in agreement with the result of Figure 10a at R = 20 kΩ. Furthermore, Figure 12 presents the output voltage and power as a function of the load resistances at the fourth defect band (NBG4$^{th}$) for the piezoelectric patch placed in three different positions. The positions 1, 2, and 3 are shown in Figure 12a, with the position in the y direction of the gravity center for the piezoelectric patch being located at 6.25 mm, 5.25 mm, and 4.25 mm above that of the supercell, respectively, and the position along the x direction of the gravity center adhering consistently with that of the supercell. It can be found from Figure 12b,c that the different positions of the piezoelectric patch lead to different output voltage and power. Hence, the harvesting of vibrational energy using point defect modes is highly dependent on the position of the piezoelectric patch because of the distribution of the maxima and minima of the localized displacement field, where the displacement field structure of point defects is extremely complex. The different position of the localized field is distributed with different pressure, so the different positions produce a different pressure difference *p* between the two sides of the patch with different degrees of deformation when the piezoelectric patch is placed in different positions of the localized displacement field. Finally, the greater the pressure difference applied, the larger the output voltage harvested. However, the position of the piezoelectric patch does not influence the optimal load resistance corresponding to the maximum power. Because we can see from Equation (5) that the optimal load resistance is determined by vibration frequency, capacitance of the piezoelectric patch,

the piezoelectric coupling coefficient, and the damping ratio, the external stimuli do not affect the value of the optimum resistance, although only if the material parameters are not varied.

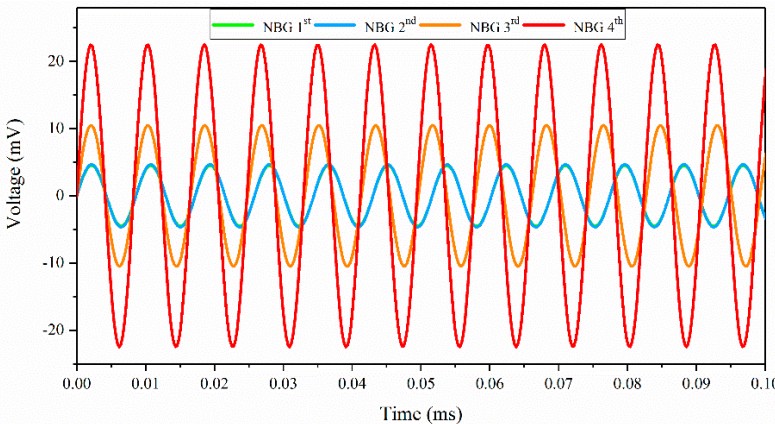

**Figure 11.** The output voltage from the piezoelectric material versus the time at defect bands of the NBG at $\sigma_z$ = 0 MPa, $H_z$ = 1 kOe, R = 20 k$\Omega$.

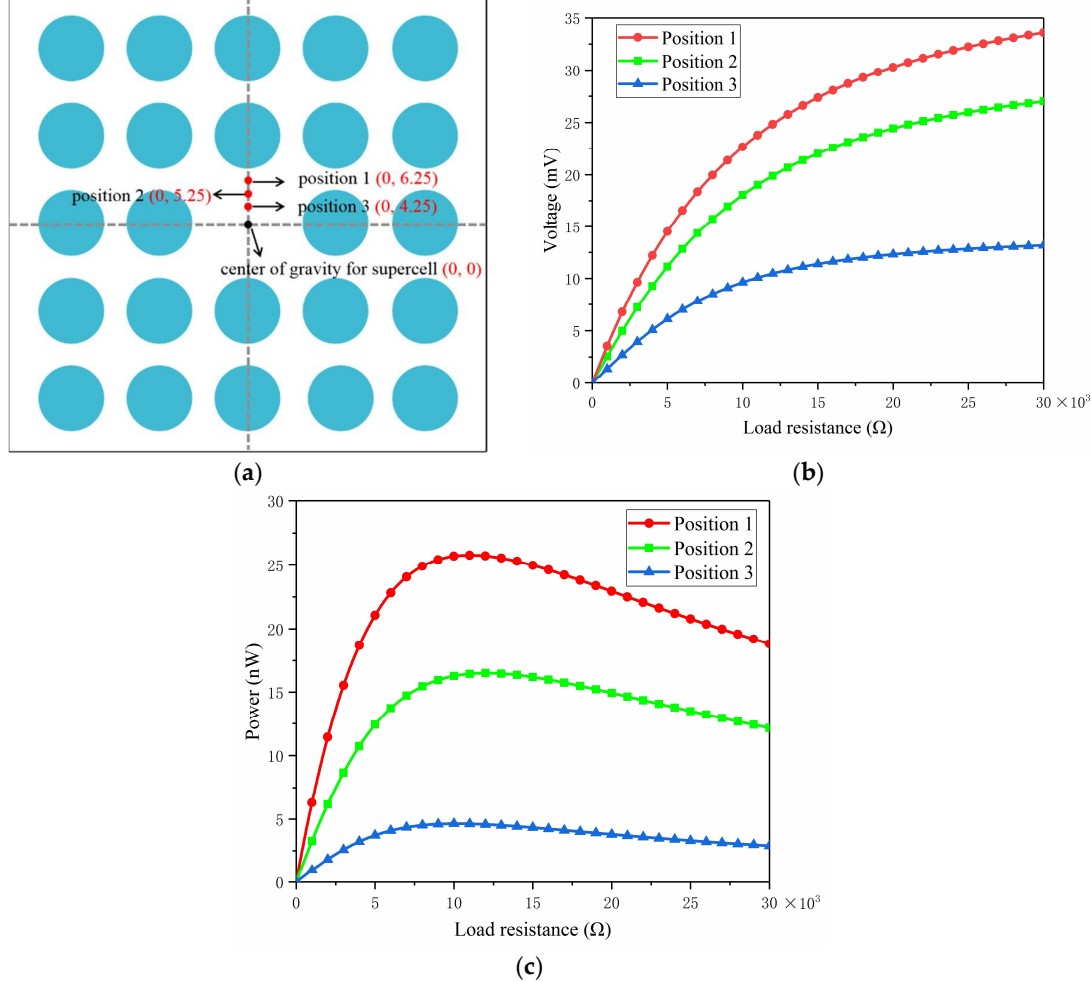

**Figure 12.** (**a**) Diagram of the piezoelectric patch placed in three different positions of the M-E PC with point defect. (**b**,**c**) show the output voltage and power of NBG4[th] versus the load resistance for different positions of the piezoelectric patch at $\sigma_z$ = 0 MPa, $H_z$ = 1 kOe.

### 3.3. Output Voltage for Anti-Plane Mode (Z Mode)

Figure 13 depicts the effects of the magnetic field and piezoelectric patch on the band structure (Z mode) of the M-E PC with point defect. It can be observed in Figure 13a that two BGs appear in the band structure and no defect band exists in the BGs of the PC without a point defect, as $\sigma_z = 0$ MPa, $H_z = 0$ kOe. It can be seen in Figure 13b that the ranges of the FBG and second band gap (SBG) are 0.333796–0.790031 and 1.128115–1.365773, respectively. When a perfect PC is transformed into a point defect PC, three defect bands are trapped in the FBG and five defect bands are trapped in the SBG as $\sigma_z = 0$ MPa, $H_z = 0$ kOe. By comparing Figure 13b,c, it can be found that the piezoelectric patch has no obvious effect on the BGs and defect bands. As magnetic field $H_z$ increases from 0 to 1 kOe, we can see from Figure 13d that the lower edge of the FBG has slightly changed, while the upper edge of the FBG decreases to 0.736598, but that the corresponding defect bands of the FBG are invariant. Note that the positions of the lower and upper edge in the SBG decline obviously but that the width of the SBG is almost unchanged and the corresponding defect bands of the SBG decline significantly.

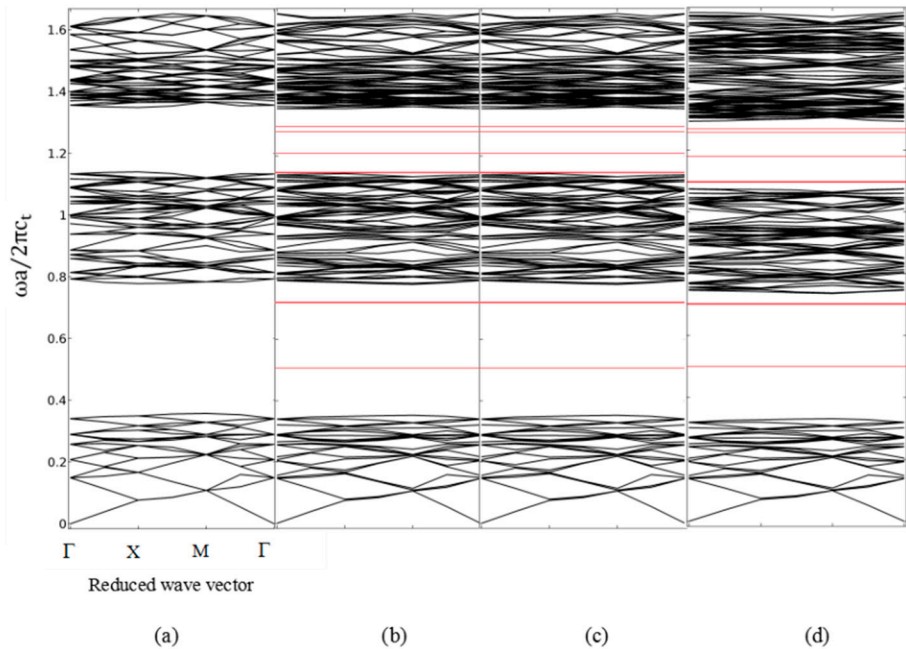

**Figure 13.** The band structure (Z mode) of the M-E PC with point defect under different magnetic fields. (**a**) perfected M-E PC at $\sigma_z = 0$ MPa, $H_z = 0$ kOe. (**b**) $\sigma_z = 0$ MPa, $H_z = 0$ kOe without PZT-5A. (**c**) $\sigma_z = 0$ MPa, $H_z = 0$ kOe with PZT-5A. (**d**) $\sigma_z = 0$ MPa, $H_z = 1$ kOe with PZT-5A.

Figure 14 presents the relation between the frequency of the defect bands and magnetic field in the M-E PC (Z mode) with point defect for the magnetic field changing from 0 to 4 kOe. The lower and upper edge FBG represent the start and cut off frequency of first band gap, respectively. Three defect bands trapped in the FBG are expressed as the 1st, 2nd, and 3rd respectively. The SBG and those corresponding to the defect bands have the same representation. We can easily find that the upper edge of the FBG declines clearly and then tends to a stable value at the saturated magnetic field, but that the lower edge FBG is unchanged with a rise in $H_z$, which leads to the width of the FBG decreasing in the low and intermediate magnetic field. Differently from the case of the *XY* modes, the defect bands of the FBG for the Z mode are nearly unchanged, and new BG cannot be opened up as $H_z$ increases. It is noteworthy that the positions of the lower and upper edges of the SBG and the corresponding defect bands decrease dramatically in the low and intermediate magnetic field and then tend to remain constant in the high field. The physical mechanism of these phenomena is determined by the certain effective material parameters of magnetostrictive material given in Ref. [25], where those parameters

show a complicated variation when the magnetic field is in the low and intermediate field, and then tend to a fixed value in the high field.

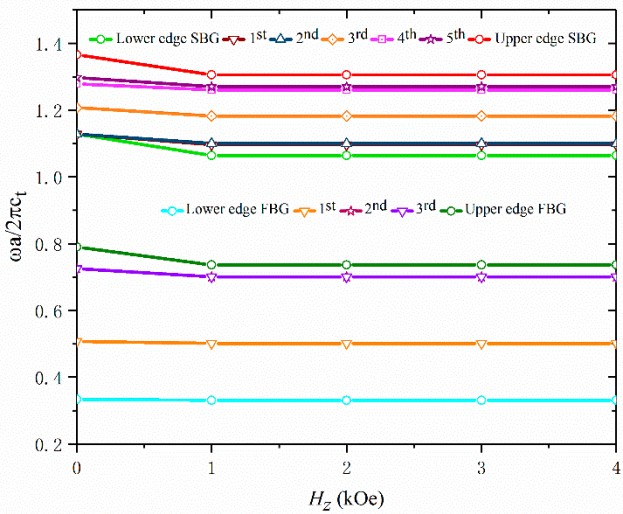

**Figure 14.** Variation in defect band frequency (Z mode) as a function of the magnetic field with given $\sigma_z = 0$ MPa.

Figure 15 presents the output voltage of the defect bands (Z mode) as a function of the magnetic field when $\sigma_z = 0$ MPa, R = 20 kΩ. The output voltages at the defect bands of the FBG are expressed as FBG1$^{st}$, FBG2$^{nd}$, and FBG3$^{rd}$, respectively. Output voltages at the defect bands of the SBG are expressed as SBG1$^{st}$, SBG2$^{nd}$, SBG3$^{rd}$, SBG4$^{th}$, and SBG5$^{th}$, respectively. The acceleration excitation agrees with the *XY* modes. One can find that the output voltage does not change significantly with the increasing magnetic field due to a slight effect of the magnetic field on the defect bands. It can be seen that when the Terfenol-D is not applied to the magnetic field, the piezoelectric patch can collect the highest voltage of 19.2 mV in SBG5$^{th}$, meaning the output voltages of FBG3$^{rd}$ and SBG4$^{th}$ are 3.4 mV and 7.2 mV, respectively. The output voltages of other defect bands are less than 1mV. The output voltage corresponding to the defect bands rapidly decreases in the low and intermediate magnetic fields and finally remains at a stable value in the high field. Hence, we can conclude that the SBG5$^{th}$ for the Z mode harvests the highest voltage of 19.2 mV at $\sigma_z = 0$ MPa, $H_z = 0$ kOe. Because the magnetic field has very little effect on the output voltage of the Z mode and the maximum energy is harvested when the magnetic field is not applied on the Terfenol-D rod, increasing efforts have been expended on the vibration energy harvesting of the point-defected M-E PC in *XY* modes by applying different magnetic fields and pre-stress values.

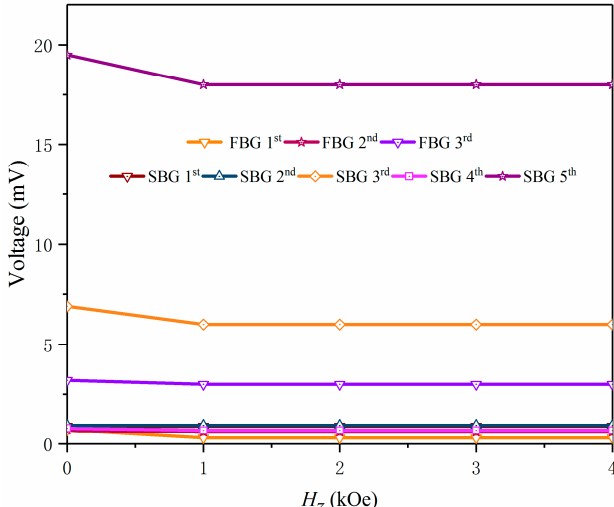

**Figure 15.** The output voltage of defect bands (Z mode) versus the magnetic field when $\sigma_z$ = 0 MPa, R = 20 kΩ.

## 4. Conclusions

In summary, a tunable vibration energy harvesting generator based on a 2D point-defected PC with magnetostrictive material and piezoelectric material under different magnetic fields and pre-stress values has been investigated. Owing to the fact that the elastic wave can be localized and enhanced in the point defect of PC, higher efficiency vibration energy conversion can be realized by attaching a piezoelectric patch to the point defect area. It has been found that the piezoelectric patch has no obvious influence on the band structure of point defect modes when BGs are in the higher region. However, the locations of the piezoelectric patch at the point defect position have a significant effect on energy harvesting. For the in-plane modes (*XY* modes), the intelligent tunability of the output voltage of the defect band is realized by applied magnetic field and pre-stress, which not only achieves vibration isolation and noise control at different frequencies, but also increases the frequency range of vibration energy harvesting simultaneously. Moreover, it is easy to expand the bandwidth of energy conversion and seek the optimal output voltage and power, which makes available higher energy harvesting efficiency. The tunability effect of the magnetic field on the output voltage of defect bands is unremarkable for the anti-plane mode (Z mode). The introduction of magnetostrictive material into a point-defected PC expands the intelligent regulation of elastic wave propagation behavior in complex multi-field environments. This tunable energy harvester used by a M-E PC provides an efficient method for active collection of broad-distributed vibration energy, which will present an important approach to apply to self-powered and low-powered devices in engineering, such as wireless sensors, medical implants, and the Internet of Things.

**Author Contributions:** Y.G. conceived of the main ideas and supervised the research. T.D. performed numerical simulations, discussed the results, and wrote the manuscript. S.Z. gave constructive advice on the calculations and revised the manuscript. Subsequently, all authors made improvements to the manuscript.

**Funding:** This work was supported by the National Natural Science Foundation of China (grant nos. 11872194 and 11572143). The costs of publishing this paper were borne by the National Natural Science Foundation of China (grant nos. 11872194 and 11572143).

**Conflicts of Interest:** The authors declare no conflict of interest.

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
