# Peer review of "A Magnetic-Dependent Vibration Energy Harvester Based on the Tunable Point Defect in 2D Magneto-Elastic Phononic Crystals"

_crystals, doi:10.3390/cryst9050261_

Round 1

Reviewer 1 Report

The authors show the potential of using magnetostrictive materials to tune the phononic bandstructure of point defects in 2D phononic crystals and enhance the performance of vibration energy –based harvesters. Although the groundwork related to 2D defects in magnetostrictive material containing phononic crystals is clearly inspired in the cited work of Gu and Jin (reference 23 in the ms), the work is innovative in the part of the applicability of the tuning phenomena and it can be of relevance in the field of energy harvesting.

My only concern is about some deficiencies in the writing and I would recommend the authors to ask a native speaker for a full correction of the manuscript. Besides an improvement of the syntax and grammar, the ms needs revision of the structure of some sentences that obscures the understanding of the meaning that the authors wanted to communicate. Examples are:

[84] Wu et al. [24] the first used PVDF beam to…

[95] the researches of the vibration energy harvesting with point defected PCs are 95only focus on the case with elastic materials, in which the frequency of defect band is single.

[128] A schematic diagram of the magnetostrictive phase can be affected by pre-stress and magnetic field is shown in Fig. 1(c), which the center of each rod is [129] as the origin of the Cartesian coordinate system, where the x-y plane is located on the cross section [130] of the rod and the z-axis is parallel to the length direction.

[193] In order to obtain the band structure of point defected PCs, considered the periodicity of the structure, the supercell system is given shown in Fig. 1(a), stress-free boundary conditions are applied [194] on the free surfaces and periodic boundary conditions based on the Bloch-Floquet theorem are [195] applied on the interfaces between the adjacent supercell system.

[201] So the defect bands of magnetic-elastic PCs when subjected different magnetic field and pre-stress are obtained.

[218] 3.1. Verification the point defect modes of elastic materials

[221] In order to validate the accuracy of the system proposed in this paper which can degenerate to the point defected PC with elastic material, it can be seen in [222] Fig. 3 that the band structure of the point defected PC made of steel rod and rubber matrix in Ref. 33. [223]

[261] These similar results can be found by Gu and Jin [23], we will carry out profoundly the study of vibration energy harvesting of M-E PC with point defect in the[262] following calculation. [263]

[444] Since the system proposed in this paper has harvested a broad-distributed vibration energy, which will demonstrate an important [445] promotion effect to provide more power for wireless sensors and MEMS in engineering.

Minor issues:

Labels in figures (6), (7), (14, and (15) are difficult to read.

Author Response

Dear reviewer,

We are very grateful for the detailed and helpful comments and suggestions made by the reviewer. We have tried our best to revise the manuscript thoroughly according to these comments and suggestions. All of the revised descriptions are marked in red. The point-by-point responses are listed in the attached file. We sincerely hope this manuscript will be finally acceptable for publication in the Crystals. Thank you very much for all your help and we are looking forward to hearing from you soon.

Reviewer 2 Report

Dear Authors.

Attached to this message you can find the report. I hope they help you to improve the article. Sincerely,

The reviewer

Author Response

(The authors gave the same response as above.)

Round 2

Reviewer 2 Report

Dear Authors.

Thant you very much for attending all my suggestions. I really expect that the article has been improved and that increase its audience and soundness. Sincerely yours,

The reviewer